



# On the estimation of vertical air velocity and detection of atmospheric turbulence from the ascent rate of balloon soundings

Hubert Luce[1], Hiroyuki Hashiguchi[2]

[1]Univ Toulon, Aix Marseille Univ., CNRS/INSU, IRD, MIO UM 110, Mediterranean Institute of Oceanography, La Garde, 83041, France
[2]Research Institute for Sustainable Humanosphere, Kyoto University, Kyoto, 611-0011, Japan

*Correspondence to*: Hubert Luce (luce@univ-tln.fr)

**Abstract.** Vertical ascent rate $V_B$ of meteorological balloons is sometimes used for retrieving vertical air velocity $W$, an important parameter for meteorological applications, but at the cost of crude hypotheses on atmospheric turbulence and without the possibility of formally validating the models from concurrent measurements. From simultaneous radar and Unmanned Aerial Vehicles (UAV) measurements of turbulent kinetic energy dissipation rates $\varepsilon$, we show that $V_B$ can be strongly affected by turbulence, even above the convective boundary layer. For "weak" turbulence (here $\varepsilon \lesssim 10^{-4} \, m^2 s^{-3}$), the fluctuations of $V_B$ were found to be fully consistent with W fluctuations measured from MU radar, indicating that an estimate of W can indeed be retrieved from $V_B$ if the free balloon lift is determined. In contrast, stronger turbulence intensity systematically implies an increase of $V_B$, not associated with an increase of W according to radar data, very likely due to the decrease of the turbulence drag coefficient of the balloon. From the statistical analysis of data gathered from 376 balloons launched every 3 hours at Bengkulu (Indonesia), positive $V_B$ disturbances, mainly observed in the troposphere, were found to be clearly associated with $Ri \lesssim 0.25$, usually indicative of turbulence, confirming the case studies. The analysis also revealed the superimposition of additional positive and negative disturbances for $Ri \lesssim 0.25$ likely due to Kelvin-Helmholtz waves in the vicinity of the turbulent layers. From these experimental evidences, we conclude that the ascent rate of meteorological balloons, with the current performance of radiosondes in terms of altitude accuracy, can potentially be used for the detection of turbulence. The presence of turbulence makes impossible the estimation of W and misinterpretations of $V_B$ fluctuations can be made if localized turbulence effects are ignored.

## 1 Introduction

The vertical ascent rates $V_B$ of meteorological balloons are mainly the combination of  the free lift and fluctuations due to vertical air velocities and variations of atmospheric turbulence drag effects. Despite their frequent use all over the world, a limited number of studies tried to extract information from $V_B$. Most of these studies focused on the estimation of the vertical air velocity because this parameter is very important for many meteorological applications (e.g. Wang et al., 2009) and for the characterization of internal gravity waves (e.g. McHugh et al., 2008). Evidence of internal gravity wave fluctuations in balloon





ascent rates was reported by Corby (1957), Reid (1972) and Lalas and Einaudi (1980). Shutts et al. (1988) and Reeder et al.
(1999) described large amplitude gravity waves in the stratosphere from the analyses of $V_B$.

However, the models or methods used for retrieving vertical air velocity from balloon ascent rates are often based on crude
assumptions about atmospheric turbulence: it is either considered as more or less uniform or neglected above the planetary
boundary layer. Johansson and Bergström (2005) estimated the height of boundary layers from $V_B$ considering that $V_B$ is

mainly affected by turbulence in convective boundary layers. In fact, the free stratified atmosphere usually reveals a "sheet
and layer" structure (e.g., Fritts et al., 2003) consisting of more or less deep layers of turbulence (a few hundred of meters)
separated by quieter and generally statically stable regions. In such conditions, turbulence intensity, often quantified by
turbulence kinetic energy dissipation rates, can vary over several orders of magnitudes with height and can reach levels similar
to those met in the convective atmospheric boundary layers (e.g. Luce et al. 2019).

In addition, most studies did not validate their estimations from concurrent measurements of vertical air velocities, making
uncertain their models and hypotheses (e.g. McHugh et al., 2008; Gallice et al., 2011). Gallice et al. (2011) proposed a model
to describe balloon ascent rates in presence of free-stream turbulence. Even if the variations of the drag coefficient with altitude
were taken into account, the intensity of turbulence was considered as uniform in the free atmosphere. Their expression of
drag coefficient was based on a mean turbulent state and thus, the model did not consider the possibility of localized layers of

turbulence, as acknowledged by the authors. Wang et al. (2009) retrieved vertical air velocity from radiosondes and dropsondes
assuming that turbulence has a negligible effect above the convective boundary layer so that the drag coefficient was
considered as nearly constant. Comparisons with wind profiler data (their Fig. 7) showed poor agreements. Most profiles
revealed oscillations, signature of gravity waves. McHugh et al. (2008) noted large (always positive) variations in balloon
ascent rate around the tropopause over Hawaii and interpreted these localized peaks as strong increases of $W$ due to mountain

waves around their critical levels. Independent measurements could not validate this interpretation and possible turbulence
effects were not considered when interpreting observations. Houchi et al. (2014) used a model similar to Wang et al.'s (2009)
model for statistical estimates of the vertical air velocity. The authors assumed that the balloon ascent rate is the sum of the
ascent rate in still air and vertical air velocity.

Modelling the ascent of balloons is not an easy task especially if the free-stream turbulence effects are not correctly taken into

account. In the present work, we studied the effects of turbulence on $V_B$ from experimental data. For this purpose, vertical
profiles of $V_B$ were compared with profiles of turbulence kinetic energy (TKE) dissipation rate $\varepsilon$ estimated from Unmanned
Aerial Vehicles (UAV) data and from the 46.5 MHz Middle and Upper atmosphere (MU) radar data. These data were gathered
during Shigaraki UAV-Radar Experiment (ShUREX) campaigns at Shigaraki MU observatory (Kantha et al., 2017). In
addition, the MU radar provided coincident estimates of vertical air velocities so that quantitative comparisons with $V_B$ could

be made. We found that a balloon is likely a good "$W$ sensor" in case of light turbulence only: under the conditions of our
experiment, $V_B$ is affected by turbulence, and thus cannot be used for estimating $W$ when $\varepsilon \gtrsim 10^{-4}\ m^2 s^{-3}$ ($1\ mWkg^{-1}$).
Therefore, a balloon is potentially more a "turbulence sensor" than a "$W$ sensor" and very large errors on $W$ can arise if the
presence of free-stream turbulence is not properly considered. Alternately, statistics on the occurrence of atmospheric



turbulence could be made from balloon ascent rates if the contribution of air motion is accurately taken into account. This

alternative purpose seems to be more achievable than retrieving $W$, except at stratospheric heights or during very calm

tropospheric conditions, as shown by earlier studies.

The effects of turbulence on the balloon ascent rate can be understood considering that this parameter in still air is given by

(Gallice et al., 2011):


$$V_z = \sqrt{\frac{8Rg}{3c_D}\left(1 - \frac{3m_{tot}}{4\pi\rho_a R^3}\right)}$$

where $R$ is the radius of the volume-equivalent sphere, g, the acceleration of gravity, $\rho_a$, the air density, and $m_{tot}$ the total

mass of the balloon, including payload, ropes, gas, etc. $c_D$ is the drag coefficient depending on the Reynolds number associated

with the balloon $Re = \rho_a V_z R/\mu$. $\mu$ is the dynamic viscosity of air. The variation of $c_D$ with $Re$ for a perfect sphere in absence

of atmospheric turbulence and for various values of turbulence intensity $Tu$ defined as the ratio of the standard deviation of

the incident air velocity fluctuations to the mean incident air velocity (e.g. Son et al. 2010) is shown in Fig. 1 of Gallice et al.

(2011). $c_D$ suddenly decreases by a factor 4 to 5 above a critical value of $Re$ (called drag crisis) so that $V_z$ can increase by a

factor 2 or more. In presence of atmospheric turbulence, the drag crisis is displaced toward lower values of $Re$ so that $c_D$ can

be reduced when crossing a turbulent layer. Recently, Söder et al. (2019) compared a profile of $Re$ with a profile of balloon

ascent rate (their figure A1) and clearly showed the existence of a drag crisis about $Re\sim4\ 10^5$ in close agreement with the

theoretical expectation for a sphere (Fig. 1 of Gallice et al. 2011). Gallice et al. (2011) proposed another (smoother) model

from experimental data with a more realistic shape of balloons and by considering heat imbalance between balloon and

atmosphere but considered a mean turbulent state of the atmosphere of $Tu\sim4\%$. This hypothesis does not hold considering the

results of comparisons we obtained.

In section 2, we briefly describe the methods used for retrieving the atmospheric parameters analyzed in the present study. In

section 3, we show comparison results between $V_B$, vertical velocity measured by MU radar, energy dissipation rate and

Richardson number profiles from three case-studies selected from ShUREX2017. These comparisons clearly indicate that

turbulence effects dominate the balloon ascent rate. The results of a statistical analysis from 376 balloons and based on the

intimate relationship between turbulence and Richardson number $Ri$ are shown in section 4. They confirm that $V_B$ is dominated

by turbulence effects when $Ri \lesssim 0.25$. Finally, conclusions of this work are given in section 5.

**2 Methods**

**2.1 Estimation of $V_B$**

200-g rubber balloons manufactured by TOTEX were equipped with RS92SGPD radiosondes for pressure, temperature,

relative humidity and horizontal wind measurements during ShUREX campaigns. Their ascent rate $V_B$ was calculated from



$\Delta z / \Delta t$ where z is the GPS altitude of the radiosondes and $\Delta t = 1\ s$. A 10-s rectangular window was applied to $V_B$ to reduce

the noise, likely due to pendulum effects, self-induced balloon motions, among other causes. For the case-studies, we focused

on the data from the ground (384 m ASL at MU Observatory) up to the altitude of 7.0 km ASL. This is primarily because (1)

the datasets were originally processed for comparisons with UAV data and UAVs did not fly above altitudes of a few km, (2)

a limited height range makes the description of individual turbulent events less tedious, (3) the increasing horizontal distance

between the radar and balloons with height due to the jet-stream becomes an important factor of uncertainty when doing

comparisons, (4) the signal-to-noise ratio (SNR) of radar measurements is statistically decreasing with height in the

troposphere and low SNR values produce additional uncertainties

## 2.2 Detection of turbulence from TKE dissipation rate $\varepsilon$

TKE dissipation rate $\varepsilon$ is a key parameter describing the intensity of dynamic turbulence. It is thus well adapted for the present

purpose, i.e. the identification of turbulent layers when the balloons were flying. $\varepsilon$ can be calculated from UAV data using two

methods described by Luce et al. (2019). A direct estimate is obtained from one dimensional (1D) spectra of streamwise wind

fluctuation measurements. An indirect estimate is deduced from temperature structure function parameter $C_T^2$ calculated from

1D temperature spectra. Similar levels of $\varepsilon$ and $\varepsilon(C_T^2)$ give credence to the results since the two estimates are independent. In

addition, consecutive profiles can be obtained during UAV ascents and descents, depending on the configuration of the flights.

Therefore, both vertical profiles of $\varepsilon$ and $\varepsilon(C_T^2)$ during ascents and descents will be shown when available.

TKE dissipation rate can also be estimated from MU radar data using the variance $\sigma^2$ of Doppler spectrum peaks produced by

turbulence. It is based on an empirical model proposed by Luce et al. (2018) and validated from comparisons with UAV-

derived $\varepsilon$. The expression of the model is $\varepsilon(MU) = \sigma^3 / L_{out}$ where $L_{out} \sim 60\ m$. In the present work, an estimate of $\varepsilon(MU)$ at

a given altitude z is obtained from an average of the values of $\sigma^2$ over +/-1 min (about 30 values since radar profiles were

obtained every ~4 sec) around the time that the altitude z was reached by the radiosonde (see also Fig. 1 of Luce et al. 2018

for a schematic). This procedure should ensure that the estimates of $\varepsilon$ are representative of those met by the balloons, assuming

horizontal homogeneity over a distance at least equal to the horizontal distance separating the balloons and the radar (up to

~30 km, see section 3). Considering that all the turbulent events analyzed in the present study persisted for more than 1 hour

and were likely associated with meso- or synoptic scale dynamics, the procedure may appear unnecessary but it is crucial for

the vertical velocity (see section 3).

Consequently, we have three independent estimates of $\varepsilon$ in the vicinity of the balloon flights. The two UAV estimates are

obtained from the ground up to ~4 km and the radar estimates in the height range 1.27-7.0 km. The radar and UAV estimates

are complementary below 1.27 and above ~4.0 km and redundant between 1.27 and ~4.0 km.



### 2.3 Estimation of vertical velocity profiles from radar data

Vertical velocities $W$ can also be directly measured from Doppler spectra when the radar beam is vertical (e.g., Röttger and

Larsen, 1990). Pseudo-vertical profiles of $W$ were reconstructed in the same way as $\varepsilon(MU)$ by averaging over +/-1 min around

the time that the altitude z was reached by the radiosonde. A two-minute averaging was applied in order to reduce the statistical

estimation errors and is suitable for detecting $W$ fluctuations of periods significantly larger than 2 minutes.

As shown by, e.g., Muschinski (1996), Worthington et al. (2001) or Yamamoto et al. (2003), $W$ can be biased by a few tens of

$cm\ s^{-1}$ or more because of refractivity-surface tilts produced by Kelvin-Helmholtz or internal gravity waves. However, this

potential bias cannot explain the large differences of a few $ms^{-1}$ between $W$ and the vertical air velocities supposed to be

deduced from $V_B$ (see section 3).

### 3 Case-studies

Three balloon flights (hereafter called V6, V14 and V16) performed during ShUREX2017 on 18 and 26 June 2017 are analyzed

in detail.  Figure 1 shows the horizontal trajectories of the balloons up to the altitude of 7.0 km ASL. The nearly circular

patterns of the UAV trajectories are also shown. The MU radar is at the position (0,0).

The balloons were intentionally underinflated with respect to standard procedures in order to get a mean ascent rate of ~2

$ms^{-1}$ similar to the vertical ascent rate of the UAVs. V6, V14 and V16 reached the altitude of 7.0 km ASL within about 33,

52 and 53 min respectively and their mean vertical ascent rates were about 3.3, 2.1 and 2.1 $ms^{-1}$. V6 drifted by less than 15

km southwestward when reaching the altitude of 7.0 km. V14 and V16 drifted by about 30 km mainly eastward due to the

influence of the sub-tropical jet-stream.

### 3.1 Analysis of the radar data

Time-height cross-sections of MU radar Doppler variance $\sigma^2$ $(m^2s^{-2})$, echo power (dB) and vertical velocity $(ms^{-1})$ around

the times of the UAV and balloon flights in the height range 1.27-7.0 km are shown in Figs. 2, 3 and 4 for V14, V16 and V6,

respectively (they are not shown in time order for ease of the description made below). The red and blue lines indicate the

altitude of the UAVs and balloons vs time, respectively. For easy reference, the most prominent and persisting turbulent layers

identified from enhanced Doppler variance (or $\varepsilon(MU)$) and UAV-derived $\varepsilon$  are labeled. The source of these layers is

sometimes recognizable from the morphology of the corresponding radar echoes in the high resolution power images. When

this is the case, the labels indicate the nature of the instabilities that gave rise to turbulence, otherwise the labels are "T1",

"T2", etc. "KHI", "MCT" and "CBL" refer to sheared flow Kelvin-Helmholtz Instability (e.g. Fukao et al., 2011), Mid-level

Cloud base Turbulence (e.g., Kudo et al., 2015), and Convective Boundary Layer, respectively. The presence of saturated air

is also indicated by the label "cloud". Note that enhanced $\sigma^2$ does not necessarily imply enhanced echoes (e.g. T1 in Fig. 2

and T2 in Fig. 4) because turbulence can sometimes produce faint echoes surrounded by enhanced echoes at their edges (e.g.,



Mc Kelley et al. 2005). The CBL in Fig. 2 is only guessed because the top CBL only slightly exceeded the altitude of the first radar gate but it was confirmed by the UAV observations.

The V14 case was characterized by weak turbulence except below ~1.3 km (CBL) and above ~5 km (MCT) (Fig. 2). The atmosphere was weakly turbulent between, but two events (T1 and T2) persisted around 2.3 km and between 4.0 and 4.5 km. The V16 case was also characterized by weak turbulence below 3.5-4.0 km and at least three well-defined layers associated with MCT and two instabilities within clouds (T2 and T3 in Fig. 3). The V6 case showed enhanced turbulence at almost all altitudes (Fig. 4) but distinct layers can be clearly noted: MCT around 5.0 km, KHI around 3.5 km (braided structures are

clearly visible around 15:00 LT) and less intense events around 2.5 km (T2) and just above the cloud base (T3). Turbulent layers (T1) detected from UAV data below 1.27 km are not indicated on the figures.

Rapid $W$ fluctuations (of period of ~1 min) are generally associated with MCT events. Nearly monochromatic oscillations of $W$ likely due to ducted gravity waves can also be noted below 2.5-3 km during V16 and V6 (Figs. 3 and 4). Their periods are about 9 and 6 min, respectively. The amplitude of $W$ did not exceed ~0.5 $ms^{-1}$ except in the MCT layer during V6 where $W$

fluctuated between +/- 2 $ms^{-1}$.

**3.2 Profile comparisons**

The results of comparisons between $V_B$ and atmospheric parameter profiles are shown for V14, V16 and V6 in Figs. 5, 6 and 7, respectively. Panels (a) show vertical velocity profiles from MU radar data and radiosondes. Panels (b) and (d) show UAV- and radar-derived $\varepsilon$ profiles in linear and logarithmic scales, respectively. Both representations are shown for ease of analysis.

Panels (c) show Richardson number $Ri = N^2/S^2$ profiles estimated from balloon data at 20 and 100 m resolution. Two vertical resolutions are used because $Ri$ is scale-dependent (Balsley et al., 2008).

The balloon ascent rate in still air $V_z$ was estimated from the difference between W and $V_B$ when turbulence was weak and the Richardson number was high. $V_z$ was found to be 1.8, 1.8 and 2.3 $ms^{-1}$ for V14, V16, V6, respectively and $V_{Bc} = V_B - V_z$ is shown in the figures. Indeed, the vertical fluctuations of $V_{Bc}$ coincide well with those of W outside the labeled turbulent layers

indicating that the variations in balloon ascent rate are dominated by the vertical air motions when turbulence is "sufficiently weak". It is particularly evident in Fig. 6 in the height range 1.3-3.8 km where the wavy fluctuations in $W$ (of ~0.5 $ms^{-1}$ in amplitude) coincide very well with those of $V_{Bc}$. Several radar estimates of W are shown for different time lags, multiple of ~9 min corresponding to the period of the wave in the radar image (Fig. 3). The fluctuations of $W$ and $V_{Bc}$ are in phase. The $W$ profile suggests that the oscillations still occurred above 3.8 km even if they were affected by the higher frequency disturbances

produced by the MCT layer around the altitude of 4.7 km (see the larger variability of the $W$ profiles). The $V_{Bc}$ profile indicates enhanced values up to +1.8 km at 5.5 km that are clearly not related to vertical air motions.

In contrast, wherever UAV- and radar-derived $\varepsilon$ estimates are enhanced in the labeled height ranges, $V_{Bc}$ is also enhanced and $V_{Bc}$ and $W$ strongly differ. Note that the UAV profiles of $\varepsilon$ during ascents and descents are very similar and there is a good agreement with the radar-derived profiles obtained during the balloon flights. Therefore, we can reasonably assume that these



profiles are representative of the turbulence conditions met by the balloons. In general, the height ranges of enhanced $\varepsilon$ coincide with minima of $Ri$, close to the critical value of 0.25, as expected for shear-generated turbulence (e.g. KHI in Fig. 7), or even less than 0, expected for MCT. $Ri$ is not necessarily small over the whole depth of the layers (e.g. around 6.0 km in Fig 5) and is surprisingly high for the whole depth of T2 in Fig. 7, but the overall results remain consistent. A puzzling result can be noted above the cloud base ($\gtrsim 6.0\ km$) during V6 (Fig. 7, as indicated by "??") where a strong increase of $V_{Bc}$ (~4

$ms^{-1}$) was neither associated with an increase of $W$ nor an increase of turbulence according to MU radar observations. A slow-down of the balloon due to precipitation loading would rather be expected. This thus remains unexplained and, by default, we must invoke horizontal inhomogeneity of $W$ and/or turbulence intensity over the horizontal distance between the radar and the balloon (~10 km). Similar features were not observed in clouds during V14 and V16.

The case-studies provided experimental evidences that turbulence can strongly increase the balloon ascent rate, very likely

through the decrease of the drag coefficient. The observed $V_{Bc}$ is thus the combination of turbulence effects and vertical air velocities. Because $W$ fluctuations appear significantly weaker than $V_{Bc}$ fluctuations, turbulence effects are likely dominant. On some occasions, increase of $V_{Bc}$ might be due to the sole turbulence effects, as in T1 of V14 (Fig. 5) since $W$ does not show any particular variations in the range of T1.

In the present cases, $\varepsilon \sim 10^{-4}\ m^2 s^{-3}$ seems to be a threshold below which turbulence does not seem to affect significantly the

balloon ascent rate. However, this value is likely specific to the present observations and may not be applicable to other conditions.

**4 Statistics**

The case-studies strongly suggest that increased balloon ascent rates are generally related to minimum values of Richardson number (negative or smaller than ~0.25 consistent with convective overturning or shear-generated instabilities in stratified

conditions, respectively). This observation can be confirmed by analyzing the relationship between $V_{Bc}$ and $Ri$ from a large amount of data. For this purpose, we used data from 376 radiosondes launched every 3 hours in Indonesia (Bengkulu, Nov-Dec 2015) during a preliminary Years of Maritime Campaign (YMC) campaign (e.g. Kinoshita et al., 2019). The choice of this dataset is arbitrary but it ensures that the same type of balloons (TOTEX-TA 200) and radiosondes (RS92SGPD) were used with similar procedures of balloon inflation for all the datasets. Figure 8 shows all the $V_B$ profiles with a slight offset for

legibility. The balloons were inflated in order to get a mean ascent rate of 5 $ms^{-1}$ (free lift). During the period of observations, the tropical tropopause layer (TTL) was often characterized by a strong temperature inversion just above the cold point temperature (CPT) around the altitude of 16~17 km (blue dots in Fig. 8) and a secondary temperature inversion of similar intensity at slightly lower altitude (red dots). For ease of statistical analysis, we refer to altitude ranges 0-16.3 km as troposphere and altitude ranges above 17.2 km (up to the top of the radiosoundings) as stratosphere.

The profiles of $V_B$ often display multiple peaks of variable widths in the troposphere especially in its upper part. In the stratosphere, the profiles are much smoother and show either weak variations or nearly monochromatic fluctuations





undoubtedly due to internal gravity waves (Tsuda et al., 1994). Therefore, we suggest that the variations of $V_B$ with height are primarily due to vertical air motions in the stratosphere and mainly due to turbulence effects in the troposphere. To confirm this hypothesis, we analyzed the relationship between $Ri$ and $V_{Bc}$ ($V_B$ corrected from the free lift). We calculated (moist)

$Ri = N_m^2/S^2$ where $N_m^2$ is the squared moist BV frequency using expression (5) of Kirschbaum and Durran (2004) at a vertical resolution of 50 m, a reasonable trade-off between 20 and 100 m used for the case-studies. Because $V_B$ seems to be weakly affected by turbulence in the stratosphere, the mean value of $V_B$ for stratospheric heights, $<V_B>_{ST}$, is expected to be a fair estimate of the ascent rate in still air ($V_z$), assuming that wave contribution is indeed removed after averaging and that other contributions are negligible. Thus, we have $V_{Bc} = V_B - <V_B>_{ST}$. $<V_B>_{ST}$ was calculated for each flight and removed

to each profile of $V_B$ in order to reduce the effects of variable mean ascent rates that may result from different balloon inflations. The mean value of $<V_B>_{ST}$ over the 376 flights was found to be precisely equal to the nominal value of $5\ ms^{-1}$.

First, the scatter plot of $V_{Bc}$ vs $Ri$ shows a very significant maximum around and below the critical value $Ri_c\sim0.25$ in the troposphere. This is an indirect confirmation that $V_{Bc}$ peaks are indeed due to turbulence (Fig. 9a), considering that small $Ri$ values are generally associated with turbulence. Second, this increase is accompanied by a larger scatter. There is no similar

tendency in the stratosphere (Fig. 9b) because $Ri$ rarely dropped below $Ri_c$, in accordance with the absence of significant turbulence guessed from the profiles of $V_B$. The variability of $V_{Bc}$ increasing with decreasing $Ri$ in Fig. 9b should mainly be due to waves.

In order to emphasize the tendency shown by Figs. 9a and 9b, averaged values of $V_{Bc}$ in $Ri$ value bands of 0.25 in width, $<V_{Bc}>$, are shown in Figs. 9c and 9d, respectively. For $Ri \gtrsim 1$, $<V_{Bc}>$ is roughly constant but slightly negative: $\sim$-0.2 $ms^{-1}$

(Fig. 9c) because $<V_B>_{ST}$ is likely not exactly the ascent rate in still air in the troposphere. This is not an important issue for the present purpose. When $Ri$ drops below $Ri_c$, $<V_{Bc}>$ increases by $\sim$+0.9 $ms^{-1}$ and remains high when $Ri < 0$ (Fig. 9a). The values for $Ri < Ri_c$ are not reliable in the stratosphere (Fig. 9d) due to the lack of data. The results shown in Fig. 9c constitute a statistical confirmation of the observations reported in section 3.

Figures 10a and 10b show $V_{Bc} - <V_{Bc}>$ vs $Ri$ for the troposphere and the stratosphere, respectively. A larger scatter is

observed around $Ri_c = 0.25$ (as emphasized by the ellipse). This cannot be explained by turbulence but likely by Kelvin-Helmholtz waves that can produce updrafts and downdrafts up to a few $ms^{-1}$ when $Ri$ reaches $Ri_c$ (see, e.g. Fukao et al., 2011). Therefore, the enhanced variability of $V_{Bc}$ when $Ri$ is small (Fig. 9a) is presumably the combination of turbulence effects and vertical air motion disturbances produced by shear flow instabilities. Assuming that the mean curve shown in Fig. 9c is statistically representative of the turbulence effects, then the scatter plot shown in Fig. 10a should also be statistically

representative of $W$ fluctuations produced by shear flow instabilities if other sources of vertical air motions are negligible.

Finally, it can be noted that the scatter plot of $V_{Bc} - <V_{Bc}>$ (Fig. 10a) is not symmetrical about 0 for $Ri > 1$ (for which turbulence is expected to be suppressed) and suggests peaks of $V_B$ (without corresponding negative disturbances) even in absence of turbulence. However, this result must be tempered by the fact that turbulence can be observed even if the estimation of $Ri$ at a given resolution is not small (see e.g., Fig. 7, T2). Measurement and estimation errors on temperature, humidity and



winds cannot be discarded on some occasions and $N_m^2$ may not be the adapted parameter for all conditions. For all these

reasons, this observation may not be indicative of more complex interactions between the balloon and the surrounding

atmosphere.

## 5 Discussion and conclusions

We found that the possibility of retrieving the vertical air velocity $W$ from radiosonde ascent rate $V_B$ highly depends on the

turbulent state of the atmosphere. In turbulent layers generated by shear or convective instabilities, $W$ cannot be measured

because $V_B$ is very likely affected by the decrease of the drag coefficient $c_D$ of the balloon. In contrast, in the calm regions of

the atmosphere, the fluctuations of $V_B$ are dominated by the fluctuations of $W$. These conditions were probably met by, e.g.,

Corby (1957), Reid (1972) and are most likely met in the lower stratosphere (Shutts et al., 1988; Reeder et al., 1999). It was

also the case during the conditions analyzed by Wang et al. (2009) above CBL. However, in light of our observations, we

speculate that Wang et al. also detected turbulent layers: localized increases of $V_B$ (up to ~$2\ ms^{-1}$) observed in the height

range 8-10 km (their Figure 1) may be attributed to turbulent layers. McHugh et al. (2008) interpreted isolated peaks of $V_B$ of

several $ms^{-1}$ of amplitude near the tropopause and at the jet-stream level in terms of $W$ disturbances around critical levels

associated with mountain waves. The absence of corresponding negative disturbances was explained by the three-dimensional

nature of the flow. Even if this interpretation is plausible, turbulence effects can be an alternative explanation since critical

levels are generally associated with turbulence. A careful scrutiny of their figures 3-7 indicates that $V_B$ increased at altitudes

where the horizontal wind shear was enhanced and temperature gradient was close to adiabatic (so that $Ri$ was likely small).

This alternative explanation is also consistent with the absence of decrease of ascent rate. Houchi et al. (2014) attributed the

spread of height increment "dz" probability density function to the sole vertical air velocity effects. Our study suggests that

part of the distribution is likely due to turbulence effects. These effects can explain upward-only motion anomaly noticed by

the authors.

It turns out that $V_B$ can also potentially be used for the detection of turbulence in the free atmosphere if the increase of $V_B$ can

be separated from the contribution of $W$. Turbulence is frequent in the free atmosphere but also very variable with height and

generally distributed in layers, especially in stratified conditions. This feature was likely not well appreciated by Gallice et al.

(2011) who considered a mean value of turbulent intensity over the whole atmosphere for establishing a model of $c_D$. The

authors themselves recognized that their model cannot work if localized turbulence –they proposed the example of turbulence

generated by gravity wave breaking- occurs.

The amplitude of the $V_B$ disturbances should depend on the variations of $c_D$ with the Reynolds number, the intensity of

turbulence and on the scales of turbulence with respect to the balloon size so that it might be difficult or even impossible to

retrieve turbulence parameters from the sole $V_B$ measurements. However, further comparisons such as shown in section 3

might be useful for establishing empirical rules on turbulence detection threshold.





*Data availability*. The balloon data are archived at the YMC Data Archive Center maintained by JAMSTEC. The radar and UAV data are still under processing for other purposes.

*Author contributions*. HL, with the help of HH, conceived of the study, carried out the analysis and retrievals, and wrote the manuscript.

*Competing interests*. The authors declare that they have no conflict of interest.

*Acknowledgments.* Radiosonde observations were carried out by JAMSTEC, BMKG, and BPPT. UAV data were provided by CU university.

*Financial support*. This study was partially supported by JSPS KAKENHI Grant Number JP15K13568 and the research grant for Mission Research on Sustainable Humanosphere from Research Institute for Sustainable Humanosphere (RISH), Kyoto
University.

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

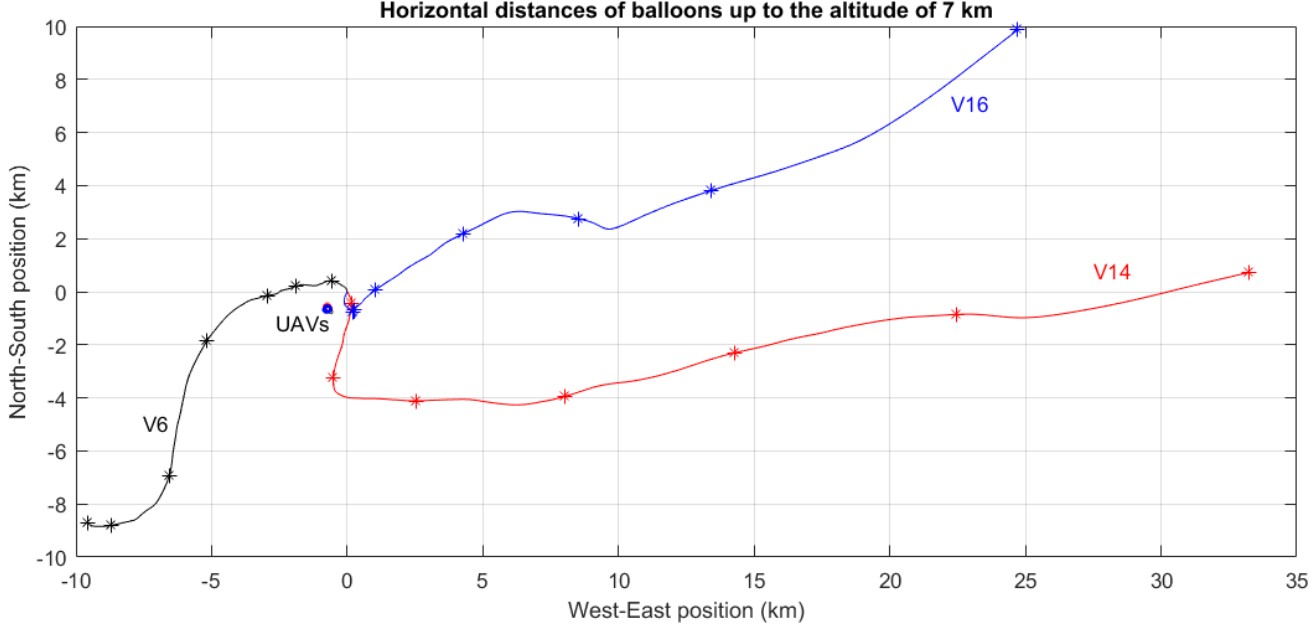

**Figure 1**. Horizontal trajectories of the meteorological balloons V6, V14 and V16. Each * symbol shows altitudes of 1 km, 2 km, etc, up to 7 km. The position (0,0) corresponds to the location of the Shigaraki MU Observatory. The circular patterns of the UAV trajectories are also shown.






**Figure 2.** (Top) Time-height cross-section of variance of the Doppler spectrum peaks corrected from the beam-broadening effects obtained from MU radar measurements during balloon flight V14 and UAV flight SH29. The altitudes of V14 and SH29 vs time are given in red and blue lines, respectively. (Middle). Same as top for radar echo power (dB) in range imaging mode. (Bottom) Same as top for vertical velocity ($ms^{-1}$). See e.g. Luce et al (2018) for more details about these figures. Labels refer to the location of turbulent layers.





**Figure 3.** Same as Fig. 2 for SH31 and V16.






**Figure 4.** Same as Fig. 2 for SH14 and V6.

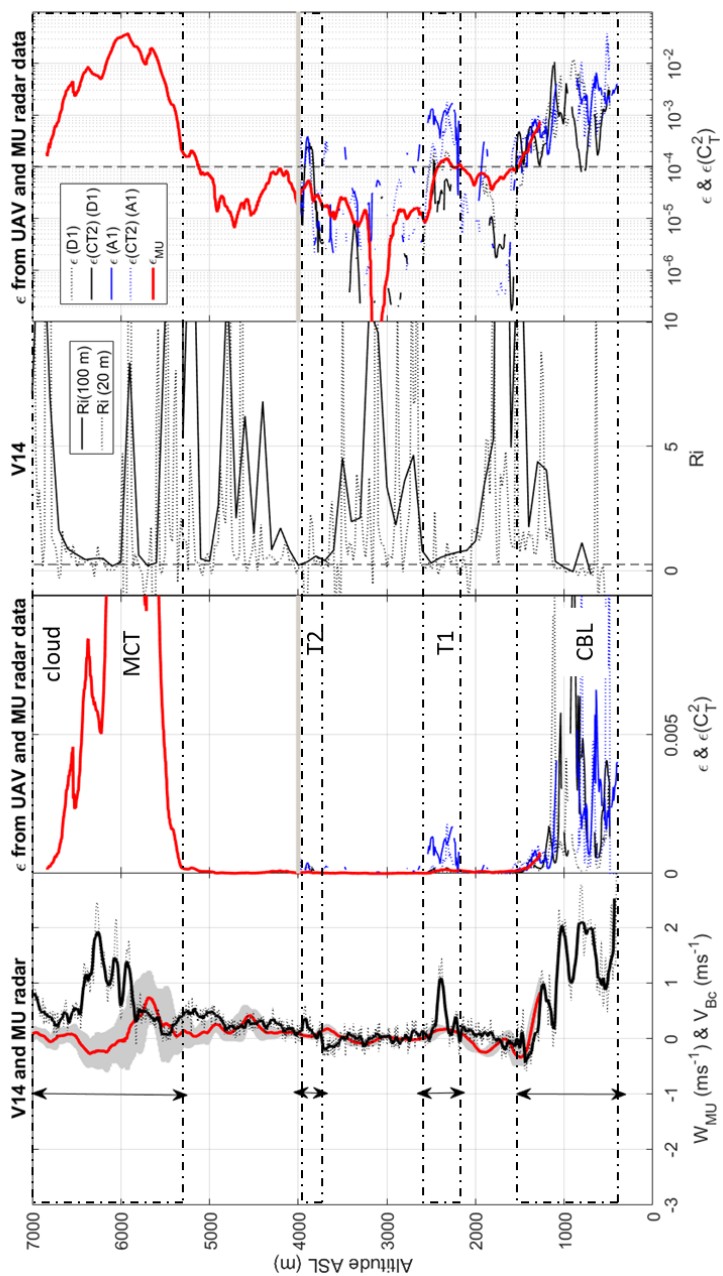

**Figure 5.** (a) Vertical profile of $V_{Bc}$ ($ms^{-1}$) (solid black: smoothed, dotted black: raw) **for V14** and $W_{MU}$ ($ms^{-1}$) (red) . The gray area shows the standard deviation of $W_{MU}$ over the averaging time (2 minutes). The vertical arrows indicate the altitude ranges affected by turbulence. (b) Vertical profiles of TKE dissipation rates $\varepsilon$ obtained from MU radar measurements (red) and UAV measurements during ascent and descent (black and blue) and using the direct and indirect methods (solid and dashed lines). The maximum altitude reached by the UAV is shown by the horizontal gray line. (c) Vertical profiles of Richardson numbers at resolution of 20 m (solid) and 100 m (dashed). The vertical dashed line indicates $Ri=0.25$. (d) Same as (b) in log scale. The vertical dashed line indicates the value of $\varepsilon = 10^{-4}\ m^2 s^{-3}$.





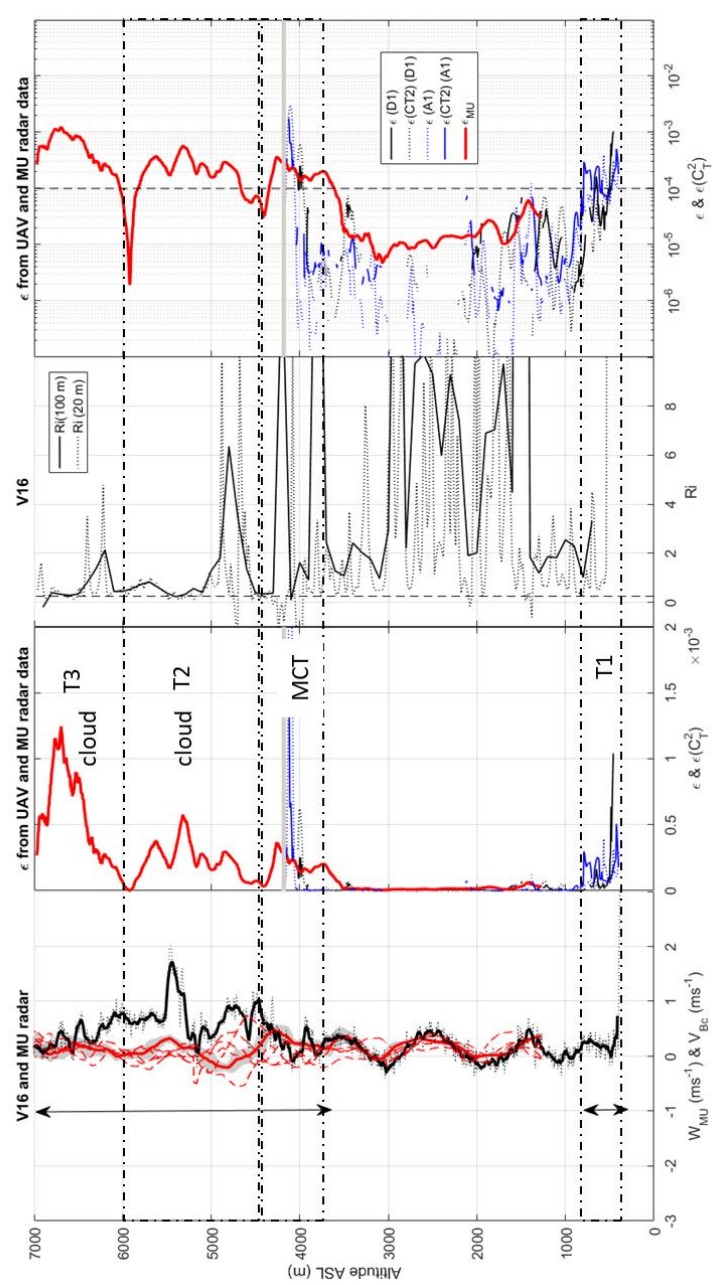

**Figure 6.** Same as Fig. 5 for V16.



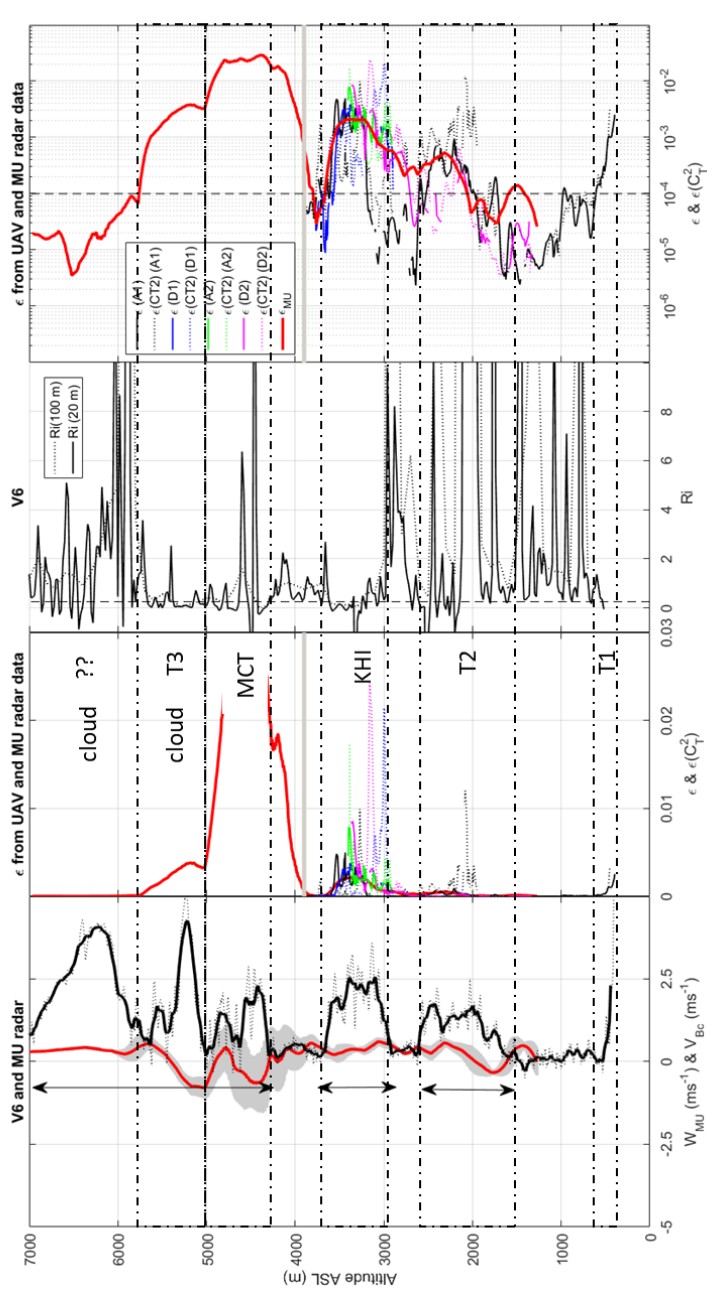


**Figure 7.** Same as Fig. 5 for V6.



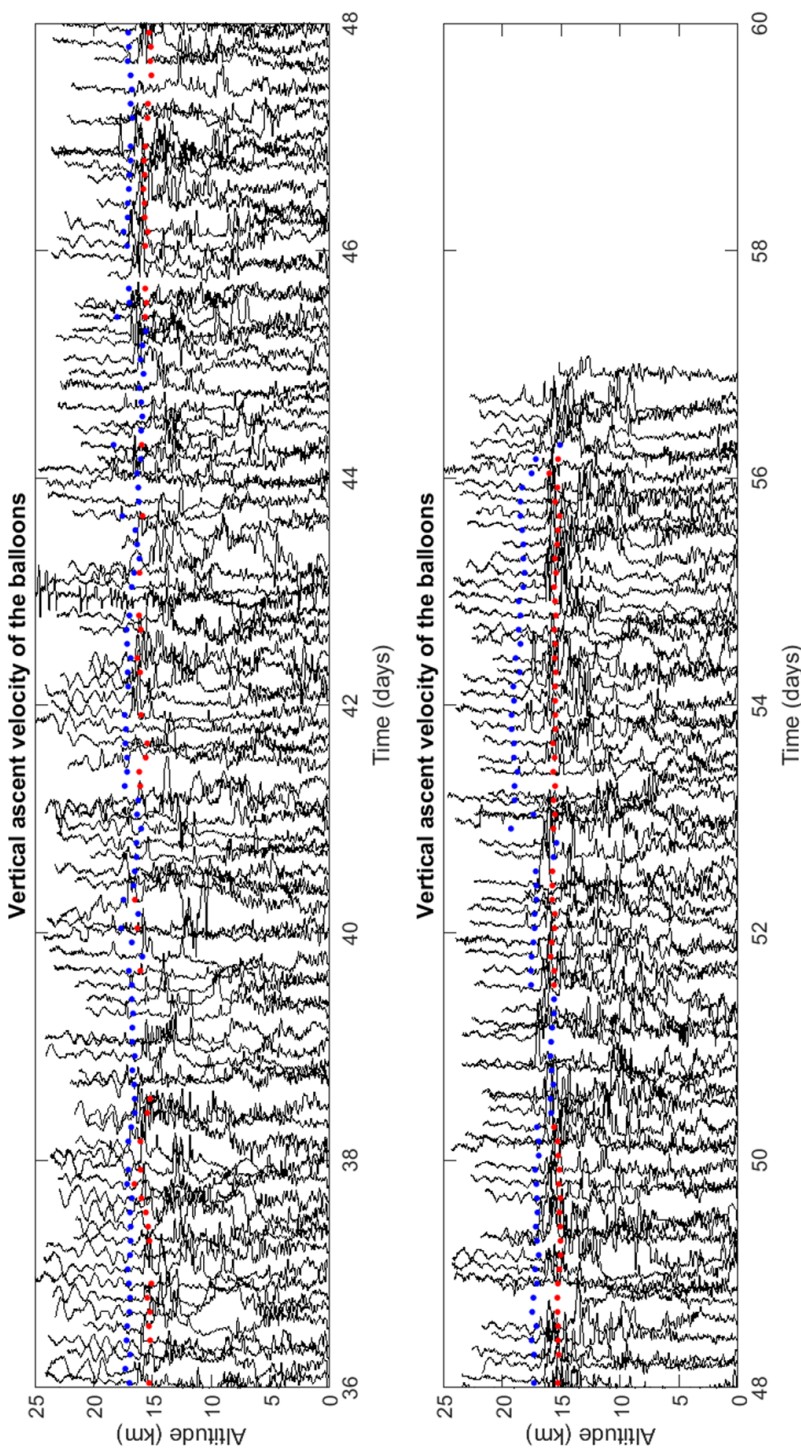

**Figure 8.** Vertical profiles of $V_B$ from 376 consecutive balloons launched every 3 hours from Nov 06 to Dec 27, 2015 during pre-YMC
campaign at Bengkulu in Indonesia.



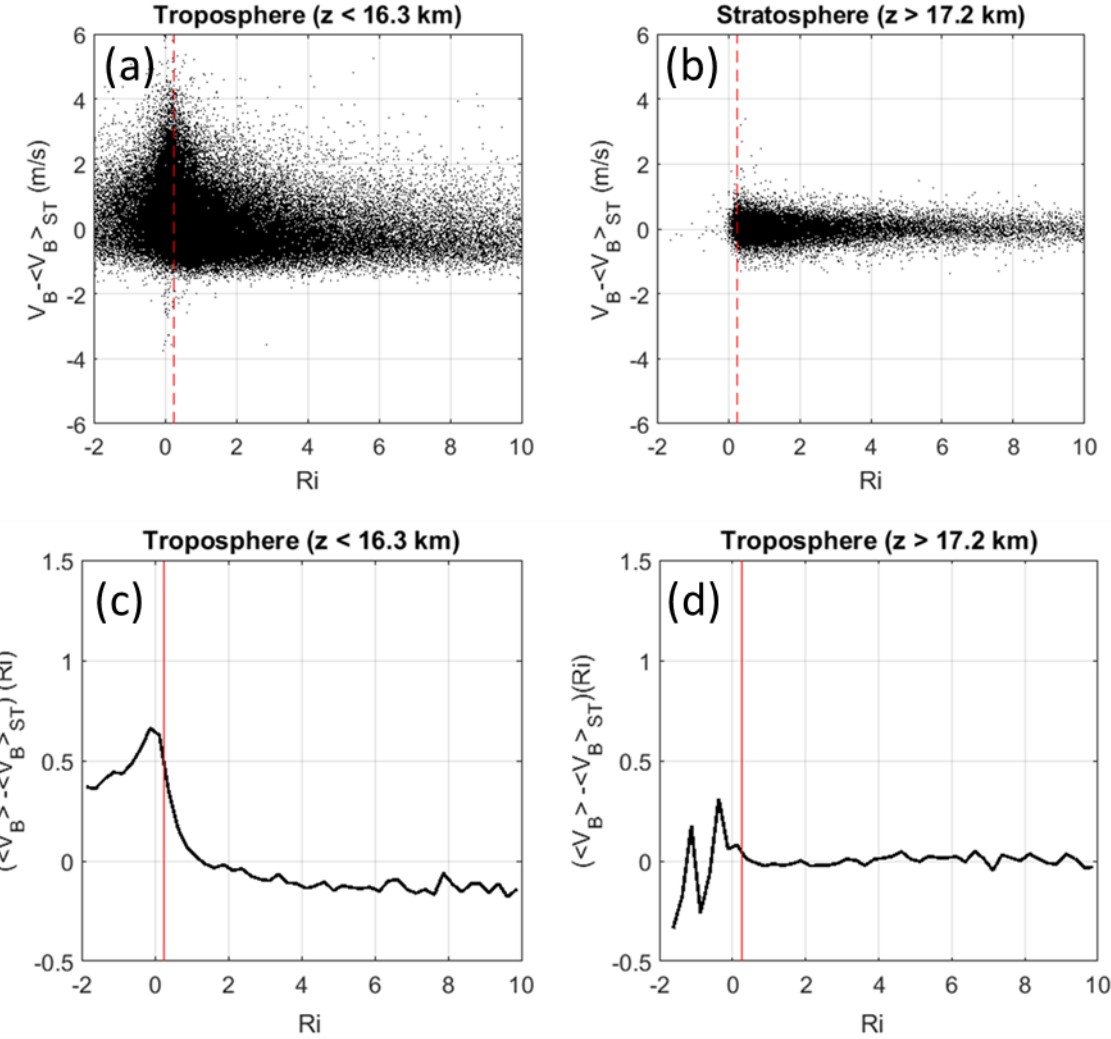

**Figure 9.** (a) Scatter plot of $V_{Bc} = V_B - \langle V_B \rangle_{ST}$ versus moist $Ri$ for the troposphere. (b) Same as (a) for the stratosphere. (c) Mean values of $V_{Bc}$ in $Ri$ bands of 0.25 in width for the troposphere. (d) Same as (c) for the stratosphere. The vertical red lines show $Ri_c = 0.25$.






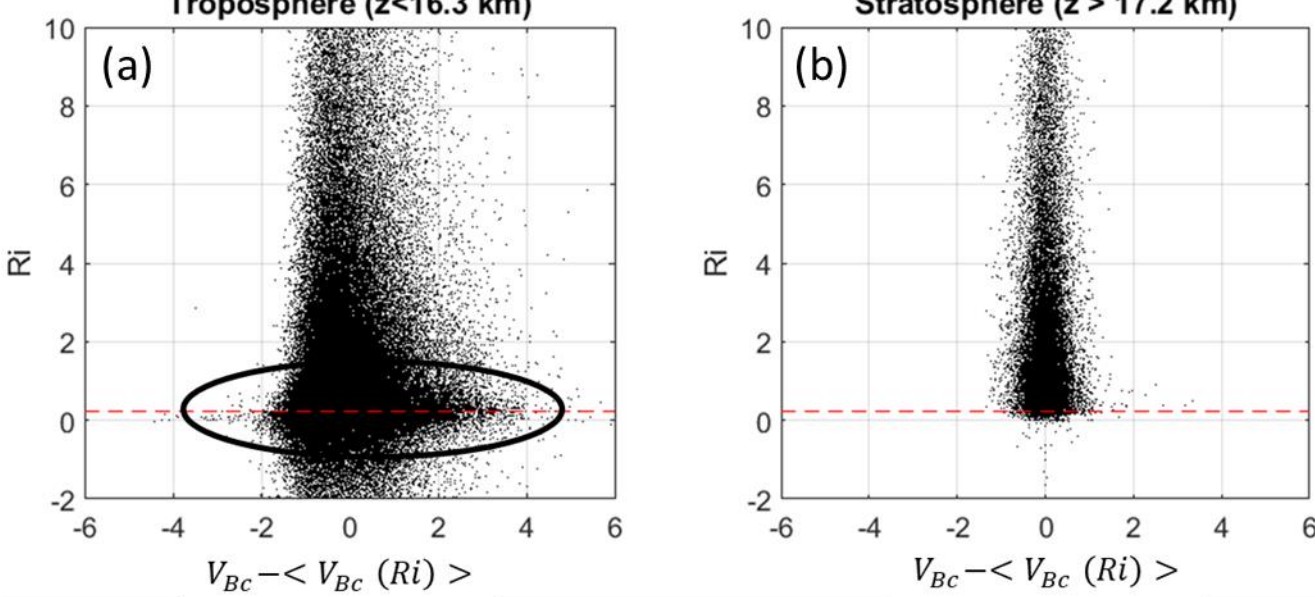

**Figure 10.** (a) Same as Fig. 9a after removing the mean tendency shown by Fig. 9c for the troposphere. (b) Same as (a) for the stratosphere. The horizontal dashed lines shows $Ri_c = 0.25$.
