# Peer review of "On the estimation of vertical air velocity and detection of atmospheric turbulence from the ascent rate of balloon soundings"

_Atmospheric Measurement Techniques, 2019_

## Referee Comment (RC1) · Anonymous Referee #1 · 24 Oct 2019

Precise measurements of vertical winds are an important topic with many meteorological applications. The paper by Luce and Hashiguchi deals with the calculation of vertical winds from radiosonde ascent rate measurements. Recently, different publications described new methods to separate the different parameters influencing the ascent rate, like vertical winds, drag coefficient of the balloon and other effects. Nevertheless, direct comparisons of retrieved vertical winds with independent observations are rare. In the first part of their paper, Luce and Hashiguchi make use of collocated UAV measurements of atmospheric turbulence and vertical wind measurements by

radar. This analysis is limited to altitudes below 7 km because the drift of the balloon and local inhomogeneities make further comparisons arbitrary. In the second part, they make a statistical analysis of a series of 376 radiosondes, confirming their results that the stability of the atmosphere influences the ascent rate of the balloon. In their main conclusion, the authors state that in a turbulent atmosphere the vertical winds can hardly be calculated without detailed knowledge of turbulence parameters. On the other hand, the ascent rate profile can be used to identify turbulence in the atmosphere. The paper is generally well written and concise. The arguments are described comprehensively and clearly. In the following, I describe only some minor comments that should be clarified before publication.

Minor comments:

ll. 82-83: I do not see the results of Gallice et al. (2011) limited to Tu=4%. The main "problem" is that they do not account for inhomogeneities in the turbulence field.

ll. 174-175: The agreement between V_Bc and W is expected from the calculation of V_z from the difference of W and V_B, and the definition of V_Bc. Is the calculation of V_z done in a different altitude than the V_Bc / W comparison?

l. 179: I am sorry, but I cannot identify the oscillations from below 3.8 km in the MCT layer above 3.8 km. Looking at the dashed lines the higher frequencies seem to dominate. Please explain.

l. 235: Please explain in short, why <V_B>_ST is not exactly the ascent rate in still air in the stratosphere.

ll. 267-270: Houchi et al. (2014) state in Section 6 a) that turbulence should broaden the ascent rate profile but not induce a tendency to purely higher ascent rates. Here, mainly the influence of turbulence on the drag coefficient is emphasized, yielding a higher ascent rate but not a broadening of the distribution. This seeming contradiction may be a question of the scales of turbulence cells. I suggest adding a clarifying

sentence.

Fig. 8: Please provide a scaling for the ascent rate and the offset.

Technical comments and typos:

l. 22: "makes the estimation of W impossible"

l. 41: "making their models and hypotheses uncertain"

l. 79: Please add a multiplication sign in "4*10ˆ5"

l. 181: "1.8 km" should read "1.8 m/s"

l. 228: ". . . in the troposphere (Fig. 9a). This is an indirect . . ."

Fig. 5-7: I suggest either to turn the figure or the figure caption by 90°.

---

## Referee Comment (RC2) · John McHugh (Referee) · 14 Nov 2019

The authors have measured the ascent rate of weather balloons, along with corresponding radar and other measurements. They point out that when atmospheric turbulence is present, the drag coefficient of the ascending balloon is reduced, and the ascent rate increases as a result. They argue that this effect is stronger that other effects, and therefore these fluctuations in ascent rate actually indicate the strength of turbulence, except in the case where turbulence is very weak. For the three case studies that they discuss, I found their arguments plausible.

They also suggest that the same arguments explain previous results, in particular the experiments by McHugh, et al (JGR, 2008), which showed an increase in ascent rate near the tropopause over Hawaii. Thus they suggest that these increases in ascent rate over Hawaii are really due to turbulence rather than a local increase in vertical velocity. However I am unconvinced that turbulence really can explain the previous results of McHugh et al. The results here show a change an ascent rate on the order of 1 m/s, but McHugh et al found an increase that was at times more than 7 m/s, meaning the balloon ascended more than twice as fast for a short distance. I am unconvinced that turbulence can cause this large of an increase. Most of this increase I think is indeed due to an increase in vertical velocity. The authors arguments don't really contradict this, as their own data only shows small increases. However I am now convinced that the large increases in ascent rate were partially due to turbulence, and thus the increase in ascent rate is overpredicting the local velocity.

I think the paper is publishable with minor revision. The revisions should include rewording the discussion of McHugh et al results with some comments about the size of the change in ascent rate.

The writing was fine. I have added a few other relatively minor issues below:

1. In figure 8, I can clearly see the difference in structure between the troposphere and stratosphere in the profiles of $V_B$, but it is not clear to me that the difference is simply waves versus turbulence, as is suggested. I think that waves are still important in the troposphere.

2. Figure 9, the 'peak' is quite broad and difficult to align with the critical $R_i$ of $0.25$ for stability. Is the breadth of this feature due to experimental error, or is the concept not quite right?

3. On page 8, '...in $R_i$ value bands of $0.25$ in width' is not an adequate description of analysis that results in figure 9c,d. What was done exactly to the data to get

this figure?

4. Why is Figure 10 rotated by 90 degrees when compared to figure 9?

5. Figures 5,6, and 7 I found to be a bit too messy, with different panels not separated by any space. It was hard to tell where one panel ended and the other began.

---

## Author Comment (AC1) · 22 Nov 2019

**Reply to Prof. McHugh**

We deeply thank Prof. McHugh for his comments. His main recommendation is to revise the discussion we proposed about the interpretation of the strong localized increases in balloon ascent rates around tropopause over Hawaii reported by McHugh et al. (2008). Our study convinced him that turbulence may have contributed to these increases but still less than upward air velocity produced by mountain waves around their critical levels.

McHugh et al.'s interpretation in terms of gravity wave effects was made plausible owing to comparisons with a mesoscale model providing evidence of mountain waves in the conditions met by the balloons. We agree that interpreting McHugh et al.'s observations in terms of turbulence effects only may be speculative (our discussion will be tempered in the revised version) but we believe that it is not unrealistic. We discuss more thoroughly this hypothesis from additional materials and examples shown below.

It should be stressed that, even if mountain activity was highlighted by McHugh et al. during the balloon flights, the vertical wind disturbances produced by the model (up to +/- 0.2 m/s, their Figures 8b and 9) were much smaller than the maximum values of ascent rate increases (a few m/s). Therefore, as noted by the authors themselves (page 8), the model did not confirm such large updrafts. In absence of independent measurements confirming or not the presence strong updrafts produced by waves, we cannot deal with the issue. However, the main arguments in favor of turbulence are:

1) All the "narrow" maxima in ascent rates (Figure 3-6 of McHugh et al.) seem to be associated with horizontal wind shears (speed and/or direction shears) and nearly adiabatic lapse rates, so that the Richardson number may be small enough for shear-generated turbulence. It would be consistent with our observations. (NB: Calculating Ri profiles for the cases shown in Figures 3-6 would be useful for confirming or not the present assertion based on a simple and inaccurate visual inspection of the figures).

2) The absence of decrease in ascent rate is consistent with a reduction of the drag coefficient due to turbulence effects. Strong three-dimensional mountain wave effects were suggested by the authors in order to overcome the absence of negative disturbances.

3) Prof. McHugh noted that the changes in ascent rate reported in our manuscript were of the order of 1 m/s (section 3) but McHugh et al. observed increases up to 7 m/s, suggesting that turbulence effects alone cannot explain the phenomenon[1].

According to Figures 1, 2 and 3 of Gallice et al. (2011), the drag coefficient $c_D$ strongly varies with the Reynolds number up to a factor ~ 4 in the range $10^{-6} - 10^{-5}$ for both idealized and experimental conditions [the drag crisis being inexistent for the experimental curves]. Then, based on these results and because $V_z \sim c_D^{-1/2}$ (expression 3 of Gallice et al.), an increase in the ascent rate by a factor up to ~2 can be predicted if *Re* strongly and quickly varies. For a standard ascent rate of 5 m/s in still air, a maximum increase of ~5 m/s is then theoretically possible. In our manuscript (section 3), we reported balloon measurements with slow ascent rates in still air (~2 m/s) because we used underinflated balloons. Therefore, ascent rate increases cannot theoretically exceed ~2 m/s. We reported ~1-1.5 m/s in Figures 5 and 6 ($V_z$ in still air was $\approx$1.8 m/s) and ~2 m/s in Figure 7 ($V_z \approx 2.3$ m/s). As a result, the differences between the changes in ascent rates reported in section 3 and in McHugh et al. can be primary due to the different ascent rates of the balloons in still air (~2 m/s and ~6 m/s[2] in our manuscript and in McHugh et al., respectively).

4) In section 4 of our manuscript, we showed scatter plots made from a large amount of balloon data from pre-YMC campaign without focusing on individual cases. In addition, Figure 8 was not clear enough for evaluating the changes in ascent rates[3]. The balloons were inflated for a standard ascent rate of 5 m/s in still air, comparable to the conditions described by McHugh et al.

Figure 1 below shows 8 consecutive profiles of balloon ascent rates $V_b$ acquired on 19 December 2015, every 3 hours from 00:00LT above the altitude of 10 km[4] and shifted by $(n-1) * 5\ m/s$ where $n$ is the flight number. For easy reference, a profile of $V_b$ shown by McHugh et al. (Figure 3) is superimposed to the profile at 06:00 LT (dashed blue line). Multiple fingers of strongly enhanced $V_b$ values can be seen below the cold point tropopause CPT (blue dots). The enhancements are typically ~2-4.5 m/s
* * *
[1] These values may depend on the method used for their estimations. Disturbances of ~1-2 m/s and ~ 3-5 m/s with respect to a "slowly varying background" can be estimated from Figures 5-7 of our manuscript, and Figures 3-6 of McHugh et al., respectively.
[2] According to Figures 3-7, it seems to be larger than 5 m/s.
[3] The figure will be corrected. In addition, the submitted figure showed half of the total balloon profiles only (by mistake)
[4] The results below 10 km are not shown for legibility of the figure.

and are thus now similar in amplitude to those reported by McHugh et al. The peaks of $V_b$ are very often associated with Richardson numbers below the critical value (altitude ranges where Ri <0.25 are indicated by the red segments). Therefore, we believe that turbulence may produce ascent rate increases similar to those reported by McHugh et al. if the balloons are inflated for an ascent rate of ~5 m/s in still air.

We now provide additional arguments suggesting that the ascent rate increases shown in Figure 1 are mainly due to turbulence effects and are not the signature of updrafts produced by gravity waves. Except maybe at 12:00 LT, $V_b$ was systematically enhanced between CPT and a secondary strong temperature inversion below (indicated by red dots). This systematic increase is recognizable in the mean profile of $V_b$ in the height range 15-16 km (thick solid line on the right side of the figure). During the campaign, the 47 MHz Stratosphere-Troposphere (ST) Equatorial Atmosphere Radar (EAR) was operating at Kototabang (Indonesia), located about 450 km North-West from the balloon launching site (Bengkulu). EAR provides similar information as MU radar with a time resolution of about 3 min and a range resolution of 150 m. Time and range resolutions of ST radars are very well adapted for studying horizontal and vertical wind disturbances produced by mountain waves (e.g. Röttger, *ST radar observations of atmospheric waves over mountainous areas: a review, A*nn. Geophys, 18, 750-756, 2000) and internal gravity waves in general. Therefore, strong gravity wave disturbances as those suggested by McHugh et al. can be detected by EAR and the interpretation of the increase of $V_b$ in terms of vertical air motions produced by waves can be tested.

Figure 2 shows time-height cross-sections of Signal to Noise Ratio SNR (dB), Doppler variance $\sigma^2 \, (m^2/s^2)$ and vertical air velocity $W$ (m/s) obtained from the vertical beam for 10 consecutive days (15-24 December 2015) in the height range 13-20 km (at time and range resolutions of 3 min and 150 m, respectively). A very persistent layer of turbulence was observed around the tropopause, between the two temperature inversions (blue and red dots) as indicated by the morphology of the SNR pattern (top panel) and, more importantly, by the persistent enhancement of Doppler variance, signature of dynamic turbulence (middle panel). The remarkable persistence of this turbulent layer (more than 10 days) and the presence of the two temperature inversions on both sides measured from balloons launched 450 km away from the radar site suggests that turbulence was produced over a very large horizontal extent and was observed at both locations. The average profile of Doppler variance $< \sigma^2 >$ (green

curve, right side of Figure 1) shows a peak at the exact location of the persistent increase in $V_b$ (15-16 km). It is thus an additional clue of the turbulent origin of the increase of $V_b$. In addition, the measurements of $W$ (bottom panel of Figure 2) do not exhibit values larger than +/-0.5 m/s; the profile of $W$ averaged over 1 day on 19 December (thick blue line, $\langle W_{EAR} \rangle$, right side of Fig.1) is associated with small standard deviations (thin horizontal blue lines) *indicating a weak wave activity*. Thus, the large increases in $V_b$ of a few m/s around the tropopause cannot be attributed to waves.

These conclusions apply to the present data set and do not necessarily fit McHugh et al.'s observations but we believe that turbulence effects only may be enough for interpreting most part of the ascent increases reported by McHugh et al. It is an alternative interpretation, not a decisive conclusion refuting wave disturbances in the conditions described by the authors.

[Figure]

Figure 1. A series of 8 consecutive profiles of $V_B$ obtained on 19 December 2019 at Bengkulu (Indonesia). (See text for more details).

[Figure]

Figure 2: Equatorial Atmosphere Radar measurements of SNR at vertical incidence (top), Doppler variance (center) and vertical velocity (bottom)

---

## Short Comment (SC1) · 25 Nov 2019

I have uploaded my comment as a supplement.

Please also note the supplement to this comment:
https://www.atmos-meas-tech-discuss.net/amt-2019-357/amt-2019-357-SC1-supplement.pdf

---

## Author Comment (AC2) · 16 Dec 2019

uploaded as a supplement

Please also note the supplement to this comment:
https://www.atmos-meas-tech-discuss.net/amt-2019-357/amt-2019-357-AC2-supplement.pdf

---

## Author Comment (AC3) · 16 Dec 2019

**Reply to Reviewer 2, Prof. J. McHugh**

**John McHugh (Referee)**

john.mchugh@unh.edu

The authors have measured the ascent rate of weather balloons, along with corresponding radar and other measurements. They point out that when atmospheric turbulence is present, the drag coefficient of the ascending balloon is reduced, and the ascent rate increases as a result. They argue that this effect is stronger that other effects, and therefore these fluctuations in ascent rate actually indicate the strength of turbulence, except in the case where turbulence is very weak. For the three case studies that they discuss, I found their arguments plausible.

They also suggest that the same arguments explain previous results, in particular the experiments by McHugh, et al (JGR, 2008), which showed an increase in ascent rate near the tropopause over Hawaii. Thus they suggest that these increases in ascent rate over Hawaii are really due to turbulence rather than a local increase in vertical velocity. However I am unconvinced that turbulence really can explain the previous results of McHugh et al. The results here show a change an ascent rate on the order of 1 m/s, but McHugh et al found an increase that was at times more than 7 m/s, meaning the balloon ascended more than twice as fast for a short distance. I am unconvinced that turbulence can cause this large of an increase. Most of this increase I think is indeed due to an increase in vertical velocity. The authors arguments don't really contradict this, as their own data only shows small increases. However I am now convinced that the large increases in ascent rate were partially due to turbulence, and thus the increase in ascent rate is overpredicting the local velocity.

I think the paper is publishable with minor revision. The revisions should include rewording the discussion of McHugh et al results with some comments about the size of the change in ascent rate.

We deeply thank Prof. McHugh for his comments. His main recommendation is to revise the discussion we proposed about the interpretation of the strong localized increases in balloon ascent rates around tropopause over Hawaii reported by McHugh et al. (2008). Our study convinced him that turbulence may have contributed to these increases but still less than upward air velocity produced by mountain waves around their critical levels.

McHugh et al.'s interpretation in terms of gravity wave effects was made plausible owing to comparisons with a mesoscale model providing evidence of mountain waves in the conditions met by the balloons. We agree that interpreting McHugh et al.'s observations in terms of turbulence effects only may be speculative but we believe that it is not unrealistic. We discuss more thoroughly this hypothesis from additional materials and examples shown below.

The discussion about McHugh et al. results (lines 261-267) has been rewritten as follows:

"McHugh et al. (2008) interpreted isolated peaks of $V_B$ of several $ms^{-1}$ of amplitude near the tropopause and at the jet-stream level in terms of $W$ disturbances around critical levels associated

with mountain waves. The absence of corresponding negative disturbances was explained by the three-dimensional nature of the flow. Even though our hypothesis remains speculative in absence of additional and independent measurements of vertical air velocity, we suggest that turbulence effects may have also contributed to the observed increase in ascent rates since critical levels are generally associated with turbulence. A careful scrutiny of their figures 3-7 indicates that $V_B$ increased at altitudes where the horizontal wind shear was enhanced and temperature gradient was close to adiabatic (so that $Ri$ was likely small)."

It must be noted that, even if mountain activity was highlighted by McHugh et al. during the balloon flights, the vertical wind disturbances produced by the model (up to +/- 0.2 m/s, their Figures 8b and 9) were much smaller than the maximum values of ascent rate increases (a few m/s). Therefore, as noted by the authors themselves (page 8), the model did not confirm such large updrafts. In absence of independent measurements confirming or not the presence strong updrafts produced by waves, we cannot deal with the issue. However, as partly stated in the manuscript, the main arguments in favor of turbulence are:

1) all the "narrow" maxima in ascent rates (Figure 3-6 of McHugh et al.) seem to be associated with horizontal wind shears (speed and/or direction shears) and nearly adiabatic lapse rates, so that the Richardson number may be small enough for shear-generated turbulence. It would be consistent with our observations. Because this assertion is based on a simple and inaccurate visual inspection of the figures, calculating Ri profiles for the cases shown in Figures 3-6 would be useful for a possible confirmation.

2) The absence of decrease in ascent rate is consistent with a reduction of the drag coefficient due to turbulence effects. Strong three-dimensional mountain wave effects were suggested by the authors in order to overcome the absence of negative disturbances.

3) Prof. McHugh noted that the changes in ascent rate reported in our manuscript were ~1 m/s (section 3 and section 4, statistics) but McHugh et al. observed increases up to 7 m/s, suggesting that turbulence effects alone cannot explain the phenomenon[1].

   a) The value of ~1 m/s is approximately the mean value obtained from statistics (Figure 9c). It means that it can be larger on many occasions. The distribution (scatter) obtained after removing the mean value is positively skewed (especially around Ri~0.25, Figure 10), possibly indicating a remaining contribution from turbulence effects.

   b) According to Figures 1, 2 and 3 of Gallice et al. (2011), the drag coefficient $c_D$ strongly varies with the Reynolds number up to a factor ~ 4 in the range $10^{-6} - 10^{-5}$ for both idealized and experimental conditions [the drag crisis being inexistent for the experimental curves]. Then, based on these results and because $V_z \sim c_D^{-1/2}$ (expression 3 of Gallice et al.), an increase in the ascent rate by a factor up to ~2 can be predicted if $Re$ strongly and quickly varies. For a standard ascent rate of 5 m/s in still air, a maximum increase of ~5 m/s is then theoretically possible. In our manuscript (section 3), we reported balloon measurements with slow ascent rates in still air (~2 m/s) because we used underinflated balloons. Therefore, ascent rate increases cannot theoretically exceed ~2 m/s. We reported ~1-1.5 m/s in Figures 5 and 6 ($V_z$ in still air was $\approx 1.8$ m/s) and ~2 m/s in Figure 7 ($V_z \approx 2.3$ m/s). As a result, the differences between the changes in ascent rates reported in section 3 and
* * *
[1] These values may depend on the method used for their estimations. Disturbances of ~1-2 m/s and ~ 3-5 m/s with respect to a "slowly varying background" can be estimated from Figures 5-7 of our manuscript, and Figures 3-6 of McHugh et al., respectively.

in McHugh et al. can be primary due to the different ascent rates of the balloons in still air (~2 m/s and ~6 m/s[2] in our manuscript and in McHugh et al., respectively).

4) In section 4 of our manuscript, we showed scatter plots made from a large amount of balloon data from pre-YMC campaign without focusing on individual cases. In addition, Figure 8 was not clear enough for evaluating the changes in ascent rates[3]. The balloons were inflated for a standard ascent rate of 5 m/s in still air, comparable to the conditions described by McHugh et al.

Figure 1 below shows 8 consecutive profiles of balloon ascent rates $V_B$ acquired on 19 December 2015, every 3 hours from 00:00LT above the altitude of 10 km[4] and shifted by $(n-1)*5 \, m/s$ where $n$ is the flight number. For easy reference, a profile of $V_B$ shown by McHugh et al. (Figure 3) is superimposed to the profile at 06:00 LT (dashed blue line). Multiple fingers of strongly enhanced $V_B$ values can be seen below the cold point tropopause CPT (blue dots). The enhancements are typically ~2-4.5 m/s and are thus now similar in amplitude to those reported by McHugh et al. The peaks of $V_B$ are very often associated with Richardson numbers below the critical value (altitude ranges where Ri <0.25 are indicated by the red segments). Therefore, we feel that turbulence may produce ascent rate increases similar to those reported by McHugh et al. around the tropopause but we also agree that these observations are not sufficient for concluding that these increases are due to turbulence effects only.

We have additional arguments suggesting that the ascent rate increases shown in Figure 1 are mainly due to turbulence effects and are not the signature of updrafts produced by gravity waves. Except maybe at 12:00 LT, $V_B$ was systematically enhanced between CPT and a secondary strong temperature inversion below (indicated by red dots). This systematic increase is recognizable in the mean profile of $V_B$ in the height range 15-16 km (thick solid line on the right side of the figure). During the campaign, the 47 MHz Stratosphere-Troposphere (ST) Equatorial Atmosphere Radar (EAR) was operating at Kototabang (Indonesia), located about 450 km North-West from the balloon launching site (Bengkulu). EAR provides similar information as MU radar with a time resolution of about 3 min and a range resolution of 150 m. Time and range resolutions of ST radars are very well adapted for studying horizontal and vertical wind disturbances produced by mountain waves (e.g. Röttger, *ST radar observations of atmospheric waves over mountainous areas: a review,* Ann. Geophys, 18, 750-756, 2000) and internal gravity waves in general. Therefore, strong gravity wave disturbances as those suggested by McHugh et al. can be detected by EAR and the interpretation of the increase of $V_b$ in terms of vertical air motions produced by waves can be tested.

Figure 2 shows time-height cross-sections of Signal to Noise Ratio SNR (dB), Doppler variance $\sigma^2 \, (m^2/s^2)$ and vertical air velocity $W$ (m/s) obtained from the vertical beam for 10 consecutive days (15-24 December 2015) in the height range 13-20 km (at time and range resolutions of 3 min and 150 m, respectively). A very persistent layer of turbulence was observed around the tropopause, between the two temperature inversions (blue and red dots) as indicated by the morphology of the SNR pattern (top panel) and, more importantly, by the persistent enhancement of Doppler variance, signature of dynamic turbulence (middle panel). The remarkable persistence of this turbulent layer (more than 10 days) and the presence of the two temperature inversions
* * *
[2] According to Figures 3-7, it seems to be larger than 5 m/s if we assume that the background (minimum) velocity is mainly due to the free lift.

[3] The figure will be corrected. In addition, the submitted figure showed half of the total balloon profiles only (by mistake)

[4] see Figure A1 (response to Prof. Drager) for the full profiles.

on both sides measured from balloons launched 450 km away from the radar site suggests that turbulence was produced over a very large horizontal extent and was observed at both locations. The average profile of Doppler variance $<\sigma^2>$(green curve, right side of Figure 1) shows a peak at the exact location of the persistent increase in $V_B$ (15-16 km). It is thus an additional clue of the turbulent origin of the increase of $V_B$. In addition, the measurements of $W$ (bottom panel of Figure 2) do not exhibit values larger than +/-0.5 m/s; the profile of $W$ averaged over 1 day on 19 December (thick blue line, $\langle W_{EAR}\rangle$, right side of Fig.1) is associated with small standard deviations (thin horizontal blue lines) *indicating a weak wave activity*. Thus, the large increases in $V_B$ of a few m/s around the tropopause may not be attributed to waves.

These conclusions apply to the present data set and do not necessarily fit McHugh et al.'s observations but we believe that turbulence effects only may be enough for interpreting most part of the ascent increases reported by McHugh et al. It is an alternative interpretation, not a decisive conclusion refuting wave disturbances in the conditions described by the authors.

[Figure]

**Figure 1**. A series of 8 consecutive profiles of $V_B$ obtained on 19 December 2019 at Bengkulu (Indonesia). (See text for more details).

[Figure]

**Figure 2**: Equatorial Atmosphere Radar measurements of SNR at vertical incidence (top), Doppler variance (center) and vertical velocity (bottom)

1. In figure 8, I can clearly see the difference in structure between the troposphere and stratosphere in the profiles of $V_B$, but it is not clear to me that the difference is simply waves versus turbulence, as is suggested. I think that waves are still important in the troposphere.

Yes, we agree, but the balloon ascent rate should be the result of multiple contributions of variable intensities (wave, convection, and turbulence effects).

2. Figure 9, the 'peak' is quite broad and difficult to align with the critical $R_i$ of $0.25$ for stability. Is the breadth of this feature due to experimental error, or is the concept not quite right?

We believe that there are several factors that make the smooth transition around $Ri = 0.25$. Without being exhaustive, the dominant factors should be:

1) The Richardson number is a scale-dependent parameter, i.e. it depends on the vertical resolution of temperature and wind profiles. A coarser resolution leads to even smoother distribution.
2) $Ri < 1/4$ is a necessary condition for active turbulence, but turbulence can be sustained up to $Ri = 1$ (thus turbulence effects can be felt even for $Ri > 0.25$).
3) The Richardson number is defined as $N_m^2/S^2$ where S is the wind shear and $N_m^2$ is the square of the moist BV frequency, when air is saturated. Contrary to $N^2$ from dry air, there are various expressions of $N_m^2$ based on different models and hypotheses on hydrometeor effects. In particular, the Kirschaum and Durran (1994) model used in the present work does not consider the presence of condensed particles but should be more relevant than the BV frequency

calculated from the equivalent potential temperature. Therefore, slight biases on $N_m^2$, and thus, on $Ri$ can be expected.

Additional data quality controls (other than made by manufacturer) have not been applied to the data for this statistical analysis. Even if the balloon launches have been performed in accordance with the manufacturer's recommendations, contaminations by balloon wake, especially when wind shear is weak, cannot be excluded. In addition, unwinder problems are not uncommon: the rope length between the balloon and payload may be smaller than the recommended length (30 m) for some flights. This problem increases the risk of wake contaminations and introduces uncertainty in wind shear altitude. These effects may contribute to incorrect estimations of Ri.

3. On page 8, '...in $R_i$ value bands of $0.25$ in width' is not an adequate description of analysis that results in figure 9c,d. What was done exactly to the data to get

[this figure ?]

'averaged values of $V_{Bc}$ in $Ri$ value bands of 0.25 in width' has been replaced by:

'mean values of $V_{Bc}$ averaged over $Ri$ segments of 0.25 in width from $Ri=-2$ to $Ri=9.75$'

4. Why is Figure 10 rotated by 90 degrees when compared to figure 9?

Figure 10 is now plotted ad Figure 9 and Figure 10b has been removed (because not informative)

5. Figures 5,6, and 7 I found to be a bit too messy, with different panels not separated by any space. It was hard to tell where one panel ended and the other began.

Thick vertical lines have been added for legibility of the figures. The legends have also been removed for clarity and a description of the curves has been added in the figure captions.

---

## Author Comment (AC4) · 16 Dec 2019

**Reply to Prof. A. Drager and Prof. P. Marinescu**

*Reply to: Comment by Aryeh Drager (Colorado State University, aryeh.drager@colostate.edu) with contributions from Peter Marinescu (Colorado State University, peter.marinescu@colostate.edu)*

We thank Prof. Drager and Prof. Marinescu for their interest to the present work and for carefully reading our manuscript. We think that our main conclusion is not very different from their point of view. There is a general agreement, based on the analysis described in section 3, that balloon ascent rates can increase due to the decrease of the balloon drag coefficient by turbulence. We somewhat differ on the strength of this effect. In particular, they suggest that the statistics shown in section 4 are not dominated by turbulence but by deep convection (i.e. the positive disturbances in ascent rates for low Richardson numbers are mainly the signature of vertical updrafts in convective cells). Without providing irrefutable evidence, we propose here additional clues and arguments suggesting that turbulence effects may rather be dominant in most cases. We hope that this information will open further discussions and investigations.

*Summary:*

*This manuscript adds to a growing body of literature on the retrieval of vertical air velocity using balloon-borne, GPS-equipped radiosonde ascent rates. One unique aspect of this study is that it attempts to provide a robust independent validation of the radiosonde-derived vertical air velocity in the form of radar-derived vertical velocity from a vertically pointing middle and upper (MU) atmosphere radar. The focus of this study, however, is the effect of atmospheric turbulence on ascent rates. Atmospheric turbulence is assessed using both the MU radar and Unmanned Aerial Vehicles (UAVs). The authors' interpretation of the data is that the effects of turbulence on balloon ascent rates are comparable in magnitude to the effects of actual vertical air motions. The authors therefore conclude if the amount of turbulence and/or the turbulence's effects on ascent rates are unknown, then it is impossible to retrieve the atmospheric vertical velocity from the radiosonde ascent rate.*

*The authors also present a separate analysis of the flights of several hundred balloon-borne radiosondes from a recent field campaign. Rather than measuring atmospheric turbulence directly, the authors use the moist Richardson number (Ri)—calculated from the temperature, humidity, and horizontal wind measurements from the radiosondes—as a proxy for the likely amount of turbulence. The authors show that low Ri in the troposphere, which is associated with greater turbulence, is associated with greater (~+0.5-0.9 m s−1) balloon ascent rates. They therefore infer that the balloon ascent rates in the troposphere are affected significantly by turbulence.*

**Reply**: The range of greater ascent rates ($\sim +0.5 - 0.9 \, m \, s^{-1}$) mentioned above refers to the mean values shown in Figure 9c. In the submitted manuscript, this range was assumed to be "statistically representative of the turbulence effects" (line 243). However, it is somewhat incorrect and we removed the sentence. Indeed, the mean value $\langle V_{Bc} \rangle$ of ascent rates should include all contributions (waves, convection, large scale billows and decrease of the drag coefficient by turbulence). For example, if we assume that turbulence effects produce a mean

increase of $X$ ($> 0$) m/s and that all other contributions can be equally positive or negative (so that the mean increase due to these contributions is 0)[1], then the total mean increase will be less than X m/s. Therefore, we believe that ascent rate increase due to turbulence effects may be significantly larger than +0.5-0.9 m/s on some occasions.

From comparisons with vertical velocities measured by MU radar (section 3), we reported ascent rate increases in stratified and clear air conditions larger than $+0.5 - 0.9 \ m/s$: namely, ~1-1.5 m/s (Figs. 5 and 6) and ~2 m/s (Fig. 7). Very importantly, these values were obtained with underinflated balloons ($V_z$ in still air was estimated to be $\approx 1.8$ m/s and 2.3 m/s, respectively). Referring to Gallice et al. (2010) and references therein, the drag coefficient can vary by a factor ~4 for the expected range of Reynolds number so that ascent rate can increase by a factor ~2, i.e. the ascent rate disturbance can be as large as the value of $V_z$ in still air, but not more. Thus, for standard balloon inflation ($V_z$~5 m/s, as is the case in section 4), the disturbance can theoretically reach ~5 m/s. The detailed reply #4 to Prof. McHugh (reviewer #2) describes cases for which values up to ~ 4 m/s are plausible (Fig. 2. See also figure A1 below).

*The sort of multi-platform analysis provided by this manuscript is sorely needed in the balloon derived vertical air velocity literature, and therefore, I appreciate the authors' time and effort towards such a study. However, after a careful reading of the manuscript, I take issue with the authors' interpretation of the data, and therefore with their main conclusions. My opinion is that the data are ambiguous as to whether turbulence or vertical air motions are affecting the balloon ascent rates. These concerns are outlined below.*

*Main major comments:*

*• One of the manuscript's key conclusions (stated in, e.g., line 22 and line 255) is that vertical air velocity W cannot be estimated using the ascent rate of meteorological balloons in the presence of turbulence. This absolute statement fails to consider situations, such as turbulent updrafts and downdrafts in deep convective storms, in which vertical motions are so strong (~tens of meters per second) that the error attributable to turbulence (seemingly ~a few meters per second) may become unimportant. The conclusion that it is "impossible" to estimate W from balloons in the presence of turbulence is therefore at best (see next major comment below) only applicable to the comparatively weak vertical air motions examined in the present study. The authors should rethink this conclusion and should, at minimum, clearly state under which atmospheric conditions their conclusions are relevant (e.g., weak versus intense vertical air motions).*

**Reply**: To the authors' knowledge, vertical motions of several tens of m/s are extremely rare even if they are sometimes sources of aviation hazards. However, we quite agree with the remainder of the comment. Turbulence effects do not necessarily prevent us to detect vertical air motions from balloon ascent rates, especially if these vertical motions are very strong.

Lines 70-73, last sentence of paragraph 4 Introduction, have been modified as follows: "This alternative purpose seems to be more achievable than retrieving *W*, except at stratospheric heights and during very calm tropospheric conditions, as shown by earlier studies, and likely during deep convective storms during which strong vertical motions are expected. "
* * *
[1] It is obviously not the case, in particular if upward motion dominates due to convection. This hypothesis is made for a simple description.

The interpretation of the balloon ascent rate is indeed always ambiguous (as rightly mentioned below by the reviewer) except when the vertical wind disturbance significantly exceeds the disturbance expected from the sole turbulence effects (see the previous reply and reply #4 to reviewer 2). In principle, the *absolute* error in vertical air motion W does not depend on $W^2$ and the *relative* error decreases with W. If turbulence produces and absolute error of +2 m/s when W=+2 m/s and W=+20 m/s, then the relative error is 100 % (apparent W=+4 m/s) and 10% (apparent W=+22 m/s), respectively. The importance of the turbulence effects depends on whether absolute or relative errors must be considered for given applications.

• *One major reservation I have regarding the analyses presented in this paper is the horizontal displacement of the balloon relative to the locations of the UAV and MU radar. These displacements of >10 km are hardly negligible. How do we know that there are not major horizontal inhomogeneities in the turbulence and vertical velocity that are leading to the observed discrepancies between the radiosonde and UAV/MU radar observations?*

**Reply:** A long horizontal distance between the instruments (and a large time lag between the measurements) can be a cause of uncertainties. This is a perennial problem we tried to minimize in the present work by providing a variety of information from UAVs and radar. It must be noted that the horizontal displacement of the balloons did exceed 10 km when comparing with MU radar data above the altitude of ~5 km, but not in the range of comparisons with UAV data (the horizontal distance was less than ~10 km for the 3 cases, see Figure 1).

We have several arguments indicating that the radiosondes crossed the turbulent layers detected by UAVs and the MU radar.

    (1) From a general point of view, turbulent layers of $10^2 - 10^3$ m in depth can have a horizontal extent exceeding $10^1 - 10^3$ km in stratified conditions, likely because they are usually associated with meso- or synoptic scale sources. There is no extensive literature focusing on this specific topic but earlier observations suggest this feature (e.g. Luce, H., R. Wilson, F. Dalaudier, H. Hashiguchi, N. Nishi, Y. Shibagaki, *Study of tropospheric turbulence from radar observations and radiosonde data using Thorpe analysis*, Radio Sci., 49, 1106-1123, 2014).
According to the MU radar observations, all the layers identified by T1, T2, …, KHI, MCT (Figs 2-4) persisted for at least 1 hour or even much more. Assuming a wind advection (~5-10 m/s in the present case), their horizontal extent should have exceeded ~30 km, i.e. the maximum horizontal distance between the balloons and radar/UAVs within the range of comparisons (Fig. 1).

    (2) The comparisons between TKE dissipation rate profiles (Figs. 5-7) give extra-credence to the hypothesis that the 3 instruments detected the same turbulent layers. These profiles estimated from radar data *at the time of the balloon flights* and UAV data show reasonable agreements in shape and levels (Figs 5-7), indicating that UAVs detected turbulent events of intensities similar to those detected by the MU radar in the same altitude range and at the time of the balloon measurements despite a time lag up to about 1 hour for V16 (see Fig. 3).

The detection of the same turbulent layers by all the sensors is a necessary condition but not sufficient. On some occasions, and especially in clouds, we agree that the horizontal inhomogeneity in the vertical velocity field within a turbulent layer may potentially explain differences between W measurements. However, this hypothesis is hardly defensible from a
* * *
[2] This assertion may not be true. For example, we do not consider the response time of the balloon to the disturbances it faces. It is thus at best an approximation.

statistical point of view. From the case studies (Figs 5-7) (and some others we did not show), vertical velocities from balloon ascent rates in the turbulent layers are systematically larger than W measured by the MU radar. This tendency is not consistent with a horizontal inhomogeneity of W because the reverse observation should also happen. In addition, if positive vertical air velocities of the order of those indicated by the peaks of $V_{Bc}$ in Figs. 5-7 are, by chance, often detected by the balloon but not by the radar at the same time, they should occur at other times on a statistical basis. Time-height cross-sections of vertical velocities in Figs 2-4 do not show any positive disturbance corresponding to the levels of the peaks of $V_{Bc}$ during the observation time (1-2 hours).

*This is of particular concern given the presence of clouds. In general, the conditions that give rise to turbulence should also give rise to inhomogeneous vertical motion, so it is hardly surprising that W and VBc do not agree as well in more turbulent layers. I do not see how one can conclude that the balloon ascent rate is changing solely due to turbulence effects on the balloon rather than at least non-negligibly due to actual vertical air motions given the potential inhomogeneities. I therefore question the paper's main conclusion that turbulent effects dominate the vertical air motion effects on balloon ascent rate.*

**Reply**: Houchi et al. (2015, p1810) reported a *broadening* of the ascent rate PDF attributed to turbulence. This broadening can indeed be due to local updrafts and downdrafts produced by the "largest" turbulent billows (i.e. of dimensions significantly larger than the balloon size). The ascent rate disturbances should be equally positive or negative because they are outside the mechanism of deep convection causing stronger updrafts. On some occasions, during past field campaigns, we experienced such situations with underinflated balloons: balloons were forced to "sink" due to strong downdrafts in the vicinity of intense turbulent layers. Vertical advection by turbulent billows can thus be the cause of the broadening of the scatter plots for Ri<0.25 (Figs. 9-10), in addition to KH waves. This point is now clarified in the manuscript (as required by reviewer 1).

*• In a similar vein, I question the conclusions drawn from the statistical analysis of 376 balloon-borne radiosondes launched in Indonesia during a recent field project. The main result for this analysis is shown in Figure 9c, in which tropospheric radiosonde ascent rates are greater for low Ri than for high Ri by ~+0.5-0.9 m s−1. with a transition near the theoretical critical Ri value of 0.25. Since lower Ri is associated with enhanced atmospheric turbulence, the authors conclude that the greater values of tropospheric radiosonde ascent rate for low Ri are due to turbulence.*

**(intermediate reply to the last sentence above)**: (a) We conclude that turbulence should be the dominant factor on many occasions (see below), but we agree that it cannot be the unique contribution when convection does occur. (b) This conclusion appears quite peremptory outside its context but it is based on the results obtained in section 3: when $Ri$ was low, turbulence was directly observed and was found to be responsible for greater values of ascent rate owing to comparisons with radar data. In section 4, turbulence is not directly detectable and collocated radar data are not available[3]. A low Ri value (<0.25) is used a proxy of turbulence. Without the results described in section 3, attributing greater values of ascent rate to turbulence effects in section 4 would be largely speculative.
* * *
[3] Radar data are shown later in this document but they were not collocated. They are provided for a statistical analysis only.

*My concern here is that lower Ri also ought to be associated with stronger vertical air motions, and these motions ought to have a net positive average vertical velocity given that these strong vertical air motions are likely to be associated with convective clouds, and updrafts in tropical convective clouds are stronger than downdrafts (e.g., LeMone and Zipser, 1980).*

**Reply:** The above comments seem to combine two distinct aspects: (1) enhanced turbulence when Ri is low (2) strong updrafts associated with convection.

The range $0 < Ri < 0.25$ characterizes dynamic shear instabilities in a statically stable background and convective instabilities are associated with $Ri < 0$ ($N^2 < 0$). The largest positive disturbances occur when $Ri$ is positive, indicating that they are rather associated with turbulence produced by shear flow instabilities. A more detailed discussion with additional data is given below.

*Bengkulu, Indonesia's location near or within the ITCZ, combined with its coastal location (which creates susceptibility for sea breeze convection), seem to have made it a locus for the formation of convective clouds during the observing time period of November/December 2015 (NASA Worldview). Therefore, an alternative interpretation of the presented data is that the enhanced radiosonde ascent rates for low Ri are due to vertical air motions induced by convective clouds rather than due to turbulence.*

We do not think that it is the most probable hypothesis for the following reasons:

[Figure]

**Figure A1**. A series of 8 consecutive profiles of $V_B$ obtained on 19 December 2015 at Bengkulu.

**1. Morphology of the ascent rate profiles**

A striking feature (common to each observation day) is the presence of narrow peaks of $V_B$ in the troposphere (Fig A1). They are generally associated with low values of Ri (sometimes negative). In the altitude range 15-17 km (tropopause layer), the peaks are almost systematic and coincide with an enhanced wind shear (and low Ri) persisting for more than 10 days (not shown). Below 15 km, they randomly occur at various heights from one flight to the other (separated by 3 hours). Their interpretation in terms of updrafts produced by deep convection is unlikely: (1) $Ri$ is not low in the lower troposphere. (2) Assuming that the balloons are

embedded within large scale convective cells and are horizontally advected by the background wind, narrow spike structures can hardly be produced[4].

**2. Statistical comparisons with radar data**

During the campaign, the 47 MHz Stratosphere-Troposphere (ST) Equatorial Atmosphere Radar (EAR) was operating at Kototabang (Indonesia), located about 450 km North-West from the balloon launching site (Bengkulu). EAR provides similar information as MU radar with a time resolution of about 3 min and a range resolution of 150 m. Despite the very large distance between the EAR and balloon launching sites, we think that the radar observations can help interpret the balloon ascent rates in a statistical sense, assuming that these observations are statistically representative of the conditions met at Bengkulu.

(1) The time-height cross-section of Signal to Noise Ratios (SNR) at vertical incidence from 15 December to 24 December 2015 (10 days) generally show a layered structure, not consistent with strong convection (Fig A2). This feature is consistent with the characteristics of the temperature and humidity profiles measured at Bengkulu: Figure A3 shows the skew-T log-P diagrams for the flights shown in Figure A1 (19 December). The tropopause level was around 150 hPa. The troposphere was close to saturation or even likely saturated (with respect to ice) on some occasions in the upper troposphere but also showed a significant stratification in humidity (large variations of the dew point temperature with height). These humidity gradients, expected to be horizontally stratified, should be the main cause of the stratified echoes. More uniform echoes associated with deep convection can sometimes be observed in the afternoons.

(2) These (short) periods of convection are usually associated with enhanced Doppler variances (i.e. enhanced turbulence) (Fig. A4). Sporadic patches of enhanced variances can be observed in the whole troposphere, consistent with local decreases of Ri in the balloon profiles.

(3) The vertical velocity W measured by EAR does not show significant disturbances (up to ~+/-0.5 m/s), except during convection periods (but not more than +/-2 m/s). The disturbances are either positive or negative (Fig A5).

(4) The histogram of W values shown in Fig A5 (Fig A6 top panel) is not positively skewed, contrary to the histogram of $V_{BC}$ (Fig A6 bottom panel). The absence of skewness is not consistent with the interpretation of the large positive values of $V_B$ in terms of (real) updrafts associated with deep convection.

(NB:The other days from 08 Nov to 14 Dec show similar features (the radar echoes were sometimes contaminated by rain echoes, producing negative outliers of W)).

This information does not support the hypothesis that the balloon ascent rate disturbances are dominated by updrafts associated with deep convection and is more consistent with turbulence effects.
* * *
[4] In addition, horizontal layers of upwelling are also not physically acceptable and the shape of peaks clearly differs from the disturbances produced by waves above the altitude of 17 km. All these arguments are rather in favor of turbulence effects.

[Figure]

**Figure A2**: Time-height cross-section of Signal to Noise Ratio at vertical incidence measured by EAR. The blue and red dots show two temperature inversions associated with the tropopause (blue dots: cold point tropopause, red dots: secondary inversion at lower altitudes) measured by the radiosondes at Bengkulu (450 km away from EAR). A long-lived turbulent layer was detected by EAR between the two inversions (see also Fig A3).

[Figure]

**Figure A3**: skew-T lop-P diagram for the 8 flights shown in Fig A1. (Black: temperature, red: dew point temperature)

[Figure]

**Figure A4**: Time-height cross-section of Doppler variance produced by turbulence measured by EAR. The Doppler variance is generally used as a proxy of dynamic turbulence.

[Figure]

**Figure A5**: Time-height cross-section of Doppler velocity (m/s) measured by EAR at vertical incidence.

[Figure]

**Figure A6**: Histogram of vertical velocities measured by EAR (top) (15-24 December) and $V_{Bc}$ (all 376 flights) (bottom).

**3. Analysis of another balloon dataset at mid-latitudes**

We performed an analysis similar to the one carried out in section 4 based on 59 balloon flights made in September 2011 at Shigaraki MU observatory, Japan, (i.e. at mid latitude where deep convection can be excluded). For various reasons (mainly few data in the stratosphere), it was not possible to estimate $V_z$ for each flight accurately so that the profiles have not been detrended. Despite a smaller amount of balloons, the tendency is similar to the one shown in Figure 9 of the manuscript, i.e. an increase of $V_B$ for low Ri. Therefore, we believe that the results shown in section 4 are "universal" and are not specific to the equatorial region.

[Figure]

**Figure A6**: results similar to Figure 9 of the manuscript using 59 balloon flights performed in September 2011 at Shigaraki MU observatory (i.e. at mid-latitudes) (see text for more details).

*References:  1) LeMone, M. A., and E. J. Zipser, 1980: Cumulonimbus vertical velocity events in GATE. Part I: Diameter, intensity and mass flux. J. Atmos. Sci., 37, 2444–2457, https://doi.org/10.1175/1520-0469(1980)037<2444:CVVEIG>2.0.CO;2.                    2) https://worldview.earthdata.nasa.gov/?v=84.93837017059609,13.306520778973313,118.547 74517059609,3.0235573460266867&t=2015-11-29T18%3A26%3A21Z*

*Other substantive comments:*

 *• Regarding line 82 and 274: To the best of my knowledge, the Gallice et al. (2011) method does not assume any particular value of Tu. Rather, this method obtains drag curves (drag coefficient as a function of Reynolds number) based on smoothed observed ascent rates and thermodynamic conditions across several launches. From page 2239 of Gallice et al. (2011):"To compensate for this lack of knowledge, and since parameters other than Re, Tu and E – such as unsteadiness or turbulence intensity length scale – are also known to affect the drag coefficient (e.g. Wang et al., 2009; Neve, 1986), an attempt is made here to derive a mean experimental drag curve for sounding balloons, based on a dataset of balloon flights."*

The statement is indeed wrong and has been suppressed in the revised version. We apologize for this erroneous interpretation. The experimental drag curves described by Gallice et al. (2011) "present a qualitative shape similar to the curves by Son et al. (2010) at *Tu=6%* and *Tu=8%*." (not 4% as stated). They further indicated that their drag curves suggest a turbulence intensity (Tu) of the atmosphere of the order of 6% to 8 % (page 2241). This statement indicates that inhomogeneities in the turbulent field were not considered.

*• Line 173: You seem to be assuming that Vz is height-invariant. What is the basis for this assumption? You seem to make the same assumption later in the paper as well, in line ~223. Given the many factors that influence a rising, expanding balloon, it is unclear to me why Vz should be assumed to have a single value throughout the troposphere and lower stratosphere.*

The examples of balloon ascent rates shown in Figure A1, indicate that assuming Vz as a constant is a reasonable approximation.

• *Line ~177-178: Shouldn't using a lag equal to a multiple of the wave's period produce essentially the same result as the non-lagged profile since you are sampling at the same phase each time? I think that using lags equal to multiples of perhaps one-eighth of the wave's period would better produce the variety that you are looking to achieve here. I can think of two additional problems with using a multiple of the wave's period: (1) if the wave propagating relative to the mean flow, then there will be a Doppler effect that produces the wrong apparent period at a given point, and (2) it takes the balloon time to ascend through a given layer – therefore, the phase of the wave it samples at the bottom of the layer will differ from the phase being sampled at the top of the layer.*

We were not able to identify with certainty the type of waves associated with these nearly-monochromatic W fluctuations revealed by the radar images. In any case, the period is only apparent (it is not the intrinsic period if there is a Doppler effect). We have some indications that they can be some kind of ducted waves according to analyses we made from other data sets in similar conditions. We did not expend the analysis because it is not related to the main topic of the manuscript.